# Seroprevalence of antibodies to SARS-CoV-2 in healthcare workers: a cross-sectional study

Joseph E Ebinger,[1,2] Gregory J Botwin,[3] Christine M Albert,[1,2] Mona Alotaibi,[4] Moshe Arditi,[2,5,6] Anders H Berg,[7] Aleksandra Binek,[8] Patrick Botting,[1,2] Justyna Fert-Bober,[2] Jane C Figueiredo,[9] Jonathan D Grein,[10,11] Wohaib Hasan,[7,12] Mir Henglin,[1,2] Shehnaz K Hussain,[13] Mohit Jain,[14] Sandy Joung,[1,2] Michael Karin,[15] Elizabeth H Kim,[1,2] Dalin Li,[3] Yunxian Liu,[1,2] Eric Luong,[1,2] Dermot P B McGovern,[3] Akil Merchant,[10] Noah Merin,[16] Peggy B Miles,[17] Margo Minissian,[1,2,18] Trevor Trung Nguyen,[1,2] Koen Raedschelders,[1,2,8] Mohamad A Rashid,[1,2] Celine E Riera,[19,20] Richard V Riggs,[21] Sonia Sharma,[22] Sarah Sternbach,[2] Nancy Sun,[1,2] Warren G Tourtellotte,[7,12] Jennifer E Van Eyk,[1,8,23] Kimia Sobhani,[7] Jonathan G Braun,[7] Susan Cheng ![ORCID][1,2,23]

► Prepublication history and additional materials for this paper is available online. To view these files, please visit the journal online (http://dx.doi.org/10.1136/bmjopen-2020-043584).

JEE and GJB are joint first authors.
KS, JGB and SC are joint senior authors.

For numbered affiliations see end of article.

**Correspondence to**
Dr Susan Cheng;
susan.cheng@cshs.org,
Dr Jonathan G Braun;
Jonathan.Braun2@cshs.org and
Dr Kimia Sobhani;
Kimia.Sobhani@cshs.org

## ABSTRACT

**Objective** We sought to determine the extent of SARS-CoV-2 seroprevalence and the factors associated with seroprevalence across a diverse cohort of healthcare workers.

**Design** Observational cohort study of healthcare workers, including SARS-CoV-2 serology testing and participant questionnaires.

**Settings** A multisite healthcare delivery system located in Los Angeles County.

**Participants** A diverse and unselected population of adults (n=6062) employed in a multisite healthcare delivery system located in Los Angeles County, including individuals with direct patient contact and others with non-patient-oriented work functions.

**Main outcomes** Using Bayesian and multivariate analyses, we estimated seroprevalence and factors associated with seropositivity and antibody levels, including pre-existing demographic and clinical characteristics; potential COVID-19 illness-related exposures; and symptoms consistent with COVID-19 infection.

**Results** We observed a seroprevalence rate of 4.1%, with anosmia as the most prominently associated self-reported symptom (OR 11.04, p<0.001) in addition to fever (OR 2.02, p=0.002) and myalgias (OR 1.65, p=0.035). After adjusting for potential confounders, seroprevalence was also associated with Hispanic ethnicity (OR 1.98, p=0.001) and African-American race (OR 2.02, p=0.027) as well as contact with a COVID-19-diagnosed individual in the household (OR 5.73, p<0.001) or clinical work setting (OR 1.76, p=0.002). Importantly, African-American race and Hispanic ethnicity were associated with antibody positivity even after adjusting for personal COVID-19 diagnosis status, suggesting the contribution of unmeasured structural or societal factors.

**Conclusion and relevance** The demographic factors associated with SARS-CoV-2 seroprevalence among our healthcare workers underscore the importance of exposure

## Strengths and limitations of this study

- ► Our study was strengthened by the size and granularity of data available on participants.
- ► Our broad definition of healthcare worker, including patient-facing and non-patient-facing employees, enhanced diversity of the study and generalisability of the results.
- ► Data collected on medical history, exposures and symptoms were self-reported.
- ► Variations in the timing of prior symptom onset in relation to the immunoassay likely resulted in underestimation of seroprevalence.
- ► Additional data on the specific roles and nature of clinical care performed by healthcare workers, including roles involving nasopharyngeal or respiratory procedures, are needed for future investigations.

sources beyond the workplace. The size and diversity of our study population, combined with robust survey and modelling techniques, provide a vibrant picture of the demographic factors, exposures and symptoms that can identify individuals with susceptibility as well as potential to mount an immune response to COVID-19.

## INTRODUCTION

Amidst the ongoing global pandemic caused by SARS-CoV-2, the viral agent causing COVID-19, substantial attention[1] turned to antibody testing as an approach to understanding patterns of exposure and immunity across populations. The use and interpretation of antibody testing to assess exposure and immunity remains fraught with inconsistencies and unclear clinical correlations, in part due to a dearth of high-quality studies

among diverse participants.[2 3] Recent publications have pointed to the challenges and importance of understanding how different antibody tests for SARS-CoV-2 perform, and factors that may render one method superior to another.[4 5] Nonetheless, there remains general agreement that antibody testing offers valuable information regarding the probable extent of SARS-CoV-2 exposure, the factors associated with exposure and the potential nature and determinants of seropositive status.[6]

To that end, we conducted a study of SARS-CoV-2 antibody screening of a large, diverse and unselected population of adults employed in a multisite healthcare delivery system located in Los Angeles County, including individuals with direct patient contact and others with non-patient-oriented work functions. Recognising the range of factors that might influence antibody status in a given individual, we focused our study on estimating seroprevalence and identifying factors associated with seropositivity and relative antibody levels within the following three categories: (1) pre-existing demographic and clinical characteristics; (2) potential COVID-19 illness-related exposures; and (3) COVID-19 illness-related response variables (ie, different types of self-reported symptoms).

## METHODS
### Study sample
The sampling strategy for our study has been described previously.[7] In brief, beginning on 11 May 2020, we enrolled a total of n=6318 active employees working at multiple sites comprising the Cedars-Sinai Health System, located in the diverse metropolis of Los Angeles County, California. The Cedars-Sinai organisation includes two hospitals (Cedars-Sinai Medical Center and Marina del Rey Hospital) in addition to multiple clinics in the Cedars-Sinai Medical Delivery Network. All active employees (total n~15 000) were invited to participate in the study by providing a peripheral venous blood sample for serology testing and completing an electronic survey of questions regarding medical history, social history and work environment in addition to COVID-19-related symptoms and exposures.[8 9] For the current study, we included all participants who completed both SARS-CoV-2 antibody testing and electronic survey forms (n=6062). Survey forms collected data on pre-existing traits, exposure factors including work location and previously experienced symptoms. Work location was specified as spending most working hours in an intensive care unit (ICU) (COVID-19 or non-COVID-19 designated), non-ICU ward (COVID-19 or non-COVID-19 designated), outpatient clinic, office, work from home or other location.

### Serological assays
For all participants, EDTA plasma specimens were transported within 1 hour of phlebotomy to the Cedars-Sinai Department of Pathology and Laboratory Medicine and underwent serology testing using the Abbott Diagnostics SARS-CoV-2 IgG chemiluminescent microparticle immunoassay (Abbott Diagnostics, Abbott Park, Illinois) performed on an Abbott Diagnostics Architect ci16200 analyser. The assay reports a signal to cut-off (S/CO) ratio corresponding to the relative light units produced by the test sample compared with the relative light units produced by an assay calibrator sample. The manufacturer-recommended S/CO ratio of 1.4 was used to assign binary seropositivity status. This cut-off was validated for high specificity (ie, >99%) ~14 days after symptom onset.[10] The Abbott assay detects antibodies directed against the nucleocapsid (N) antigen of the SARS-CoV-2 virus, which assists with packaging the viral genome after replication, and achieves specificity for IgG by incorporating an anti-human IgG signal antibody. To verify local performance of the assay, we used samples obtained at our institution from 60 cases of COVID-19 (hospitalised between March and May 2020) and 178 controls that were identified based on positive or negative PCR assay (RT-qPCR assay based on A*STAR Fortitude Kit 2.0) with a time lapse between symptom onset and antibody assay of ~7–14 days. We found a sensitivity or positive per cent agreement of 88.3%, with coefficients of variation ≤1.4% for positive and negative controls.

### Statistical analyses
#### Estimates of seroprevalence
We conducted a comprehensive literature review to identify published data (through 25 June 2020) on the sensitivity and specificity of the Abbott Architect SARS-CoV-2 IgG assay, as applied in specific populations using the manufacturer's recommended thresholds. We identified a total of 15 studies assessing sensitivity in 2114 tests and 18 studies reporting specificity in 7748 tests (online supplemental tables 1 and 2); we combined this information with data from an additional independent cohort of 60 case and 178 control specimens used to assess sensitivity and specificity, respectively, within the Cedars-Sinai Department of Pathology and Laboratory Medicine. We noted that studies investigating specificity generally assessed samples collected prior to the SARS-CoV-2 pandemic whereas studies reporting sensitivity included specimens from RT-PCR-confirmed individuals (see details provided in online supplemental tables 1 and 2). We restricted our analyses to a referent cohort of tests conducted on samples from individuals who were assayed ≥7 days following symptom onset to most closely match our cohort sample characteristics and the situational context for study enrolment. Given that our study cohort included a large number, yet not the total number, of all eligible healthcare workers employed in our health system, we used the iterative proportional fitting (IPF) procedure to account for any possible sampling bias; notably, the IPF has been applied effectively in prior as well as contemporary studies related to SARS-CoV-2 exposure.[11] Accordingly, we integrated source population-level demographic data, representative of the entire Cedars-Sinai employee base, with data from our enrolled study sample and then used IPF to estimate the number of

eligible employees within each demographic category (with provided population totals considered the target, using constraints derived from our sample).[12] In addition to accounting for potential bias from sampling, we also recognised the need to account for potential bias related to the previously reported sensitivity and specificity of the antibody assay (online supplemental tables 1 and 2). Thus, in accordance with methods applied in similar seroprevalence studies,[13 14] we fit a Bayesian multilevel hierarchical logistic regression model using RStan,[15 16] including reported age, gender, race/ethnicity and site as coefficients, to model exposure probability. We then estimated the seroprevalence within each poststratified demographic category based on the averaged and weighted value of the expected number of employees within that category.

### Factors associated with seroprevalence

Prior to logistic and linear multivariable-adjusted analyses, age and IgG index were transformed by dividing by 10 for interpretability of coefficients in all models. In adjusted analyses, we compared differences between serology status (ie, antibody positive vs negative) in each variable of interest, grouped into one of three categories: (1) pre-existing demographic and clinical characteristics (eg, age, gender, ethnicity, race and self-reported medical comorbidities); (2) COVID-19-related exposures (eg, self-reported medical diagnosis of COVID-19 illness, household member with COVID-19 illness, number of people living in the home including children, type of home dwelling, and so on); and (3) COVID-19-related response variables (eg, self-reported fever, chills, dry cough, anosmia, nausea, myalgias, and so on). In multivariable-adjusted analyses, we used logistic and linear models to examine the extent to which the three categories of variables (predictors) may be associated with antibody-positive status (primary outcome) in the total sample or IgG antibody level in the subset of persons with positive antibody status (secondary outcome). Initial models were deliberately sparse, adjusting for a limited number of key covariates (eg, age, gender), and those variables with associations meeting a significance threshold of p<0.05 were advanced for inclusion in a final multivariable model along with only other variables identified as significant from the sparse regressions. A final separate logistic or linear multivariable model was constructed for each of the three categories of variables in relation to the binary outcome of seropositivity or the continuous outcome of IgG antibody level, respectively.

### Patient and public involvement

Patients and the public were not involved in the development of this study.

## RESULTS

The demographic, clinical, exposure and symptom response characteristics of the study sample are shown in table 1, by antibody test result status; the study sample included individuals whose residence spanned diverse regions across Los Angeles County (online supplemental figure 1). The overall seroprevalence was 4.1% (95% CI 3.1% to 5.7%), with higher estimates seen in younger compared with older individuals and in Hispanics compared with non-Hispanics (figure 1 and online supplemental table 3).

In multivariable-adjusted analyses of pre-existing characteristics (figure 2 and online supplemental table 4), the main factors significantly associated with greater odds of seropositive status were Hispanic ethnicity (OR 1.80, 95% CI 1.31 to 2.46, p<0.001) and African-American race (1.72, 95% CI 1.03 to 2.89, p=0.04), compared with non-Hispanic Whites. The main factors associated with lower odds of being seropositive were older age (0.81 per age decade, 95% CI 0.71 to 0.92, p=0.001) and a history of asthma (0.48, 95% CI 0.28 to 0.83, p=0.009). Among all seropositive persons, hypertension was significantly associated with higher antibody level (beta 0.12 (SE 0.04) per 10-unit increment in the IgG index, p=0.003).

In multivariable-adjusted analyses of COVID-19-related exposures (figure 3 and online supplemental table 5), the factors significantly associated with greater odds of seropositive status were having had a medical diagnosis of COVID-19 (7.78, 95% CI 5.73 to 10.56, p<0.001) and a household member previously diagnosed with COVID-19 (9.42, 95% CI 5.50 to 16.13, p<0.001), with a similar trend observed for working in a location where patients with COVID-19 are treated (1.61, 95% CI 1.18 to 2.18, p=0.002). Among seropositive individuals, having a medical diagnosis of COVID-19 was associated with higher antibody level. Notably, domestic travel, dwelling type, number of people in the home and having children or common domestic pets were not associated with either seroprevalence or antibody level in the more completely adjusted multivariable models, which can account at least partially for the effects of unmeasured confounders that are not captured in the sparser models.

In multivariable-adjusted analyses of COVID-19 response variables (figure 4 and online supplemental table 6), the strongest self-reported symptom associated with greater odds of seropositive status was anosmia (11.91, 95% CI 7.77 to 18.24, p<0.001). Other symptoms associated with the presence of antibodies included dry cough, loss of appetite and myalgias. Notably, the symptoms associated with lower odds of seropositive status included sore throat and rhinorrhoea. Dyspnoea was significantly associated with higher IgG index levels in seropositive individuals (beta 0.13 (SE 0.04), p=0.001).

Significantly predictive pre-existing characteristics, exposures and symptoms from the prior models were subsequently analysed together. In multivariable analysis, all included predictors, except for dry cough, remained significantly associated with the presence of antibodies. Predictors which remained significantly associated with higher antibody levels included hypertension (beta 0.1 (SE 0.04), p=0.007), prior COVID-19 diagnosis (beta

**Table 1** Characteristics of the study sample

| | Antibody negative n=5850 | Antibody positive n=212 |
|---|---|---|
| **Pre-existing characteristics** | | |
| Age, mean (SD) | 41.6 (12.0) | 38.5 (11.2) |
| Male gender (%) | 1876 (32) | 73 (34) |
| Hispanic ethnicity (%) | 1097 (19) | 62 (29) |
| Race (%) | | |
| Asian | 1809 (31) | 57 (27) |
| Black | 354 (6) | 18 (8) |
| White | 2938 (50) | 104 (49) |
| Other | 749 (13) | 33 (16) |
| Current smoker (%) | 99 (2) | 3 (1) |
| Current vape user (%) | 83 (1) | 4 (2) |
| Medical conditions (%) | | |
| Asthma | 733 (13) | 14 (7) |
| Autoimmune disease | 228 (4) | 4 (2) |
| Cancer | 195 (4) | 3 (1) |
| Cardiovascular | 127 (2) | 2 (1) |
| Chronic obstructive pulmonary disease | 84 (2) | 0 (0) |
| Diabetes mellitus | 371 (7) | 8 (4) |
| Hypertension | 967 (17) | 26 (13) |
| BMI, mean (SD) | 26.7 (5.6) | 26.3 (5.1) |
| Obesity, BMI≥30 (%) | 998 (23) | 32 (21) |
| **Potential COVID-19-related exposures** | | |
| Personal diagnosis of COVID-19 (%) | 530 (9) | 104 (50) |
| Household member diagnosed with COVID-19 (%) | 51 (1) | 31 (15) |
| Domestic travel since September 2019 (%) | 2127 (37) | 54 (26) |
| International travel since September 2019 (%) | 1324 (23) | 44 (21) |
| Regular contact with patients with COVID-19 (%) | 1358 (24) | 86 (41) |
| Work on a unit housing/caring for patients with COVID-19 (%) | 1600 (27) | 93 (44) |
| Type of dwelling (%) | | |
| Apartment | 2636 (46) | 93 (44) |
| House | 2914 (51) | 107 (51) |
| Other | 216 (4) | 9 (4) |
| Number of people living in the home, mean (SD) | 2.3 (1.7) | 2.4 (1.8) |
| Any persons in the home under age 18 years (%) | 1843 (32) | 65 (31) |
| Any persons in the home under age 12 years (%) | 1467 (25) | 51 (24) |
| Cats as household pets (%) | 783 (13) | 27 (13) |

Continued

**Table 1** Continued

| | Antibody negative n=5850 | Antibody positive n=212 |
|---|---|---|
| Dogs as household pets (%) | 2189 (37) | 95 (45) |
| **Potential COVID-19-related responses** | | |
| Fever (%) | 497 (9) | 87 (43) |
| Chills (%) | 683 (12) | 95 (46) |
| Headache (%) | 2061 (36) | 126 (61) |
| Conjunctivitis (%) | 162 (3) | 14 (7) |
| Anosmia (%) | 252 (4) | 107 (52) |
| Nasal congestion (%) | 1611 (28) | 104 (51) |
| Rhinorrhoea (%) | 1493 (26) | 82 (41) |
| Dry cough (%) | 1235 (22) | 108 (53) |
| Productive cough (%) | 542 (10) | 50 (25) |
| Sore throat (%) | 1368 (24) | 81 (40) |
| Chest pain (%) | 453 (8) | 45 (22) |
| Dyspnoea (%) | 604 (11) | 66 (33) |
| Anorexia (%) | 390 (7) | 78 (38) |
| Nausea (%) | 657 (12) | 52 (25) |
| Vomiting (%) | 188 (3) | 15 (8) |
| Diarrhoea (%) | 853 (15) | 59 (29) |
| Myalgias (%) | 1033 (18) | 117 (58) |
| Fatigue (%) | 1447 (25) | 135 (66) |
| Skin changes (%) | 261 (5) | 15 (8) |
| Stroke symptoms (%) | 35 (1) | 3 (2) |
| Sneezing (%) | 1863 (33) | 94 (47) |

BMI, body mass index.

0.1 (SE 0.03), p=0.001), working in a COVID-19 unit (beta 0.06 (SE 0.03), p=0.021), dyspnoea (beta 0.08 (SE 0.03), p=0.009) and nausea (beta 0.06 (SE 0.03), p=0.05) (figure 5 and online supplemental table 7) .

## DISCUSSION

In a large diverse healthcare employee cohort of over 6000 adults in Los Angeles, we observed a seroprevalence rate of 4.1% which, when accounting for published test characteristics, may range from 3.1% to 5.7%. Seroprevalence varied across demographic, clinical, exposure and symptom-based characteristics. Specifically, factors significantly associated with presence of IgG antibodies included younger age, Hispanic ethnicity and African-American race, as were exposure-related factors including the presence of either a personal or household member having a prior medical diagnosis of COVID-19. Among self-reported symptoms, anosmia was most strongly associated with the presence of antibodies, with positive associations also noted for fever, dry cough, anorexia and myalgias. The size and diversity of this study population, combined

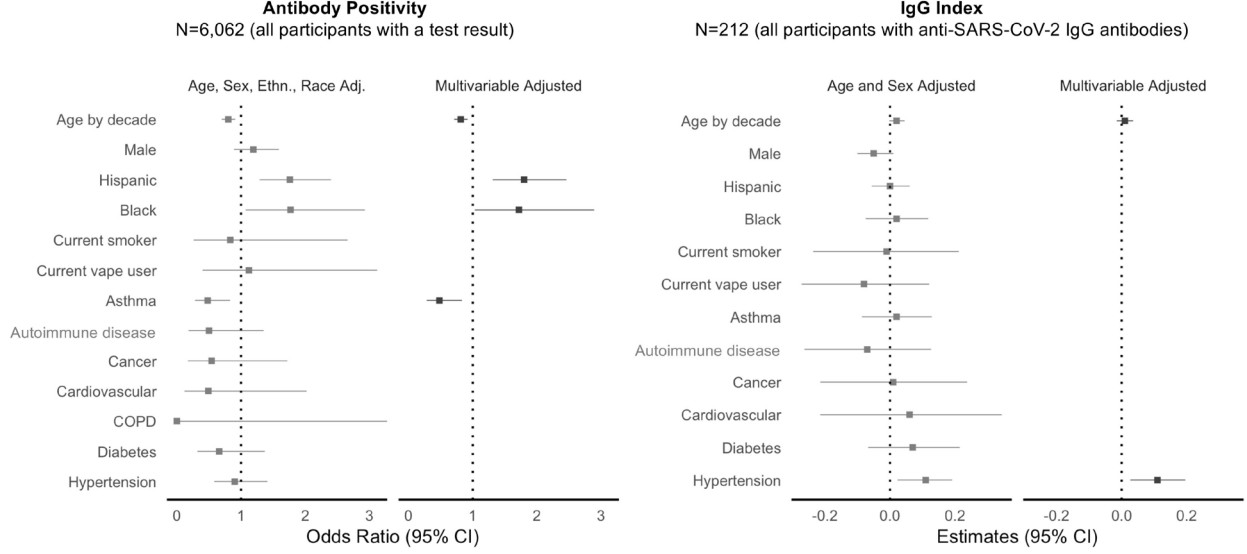

**Figure 1** Seroprevalence overall and by subgroup.

with robust survey and modelling techniques, provide a more vibrant picture of the population at highest risk for COVID-19 infection, risks of various potential exposures and symptoms that should alter patients to potential illness.

Most prior seroprevalence studies have focused on cohorts that included healthcare workers predominantly involved in direct or indirect patient care, persons living within a circumscribed region with high viral exposure rates, or larger geographic areas from which motivated

**Figure 2** Pre-existing factors associated with SARS-CoV-2 seroprevalence. COPD, chronic obstructive pulmonary disease.

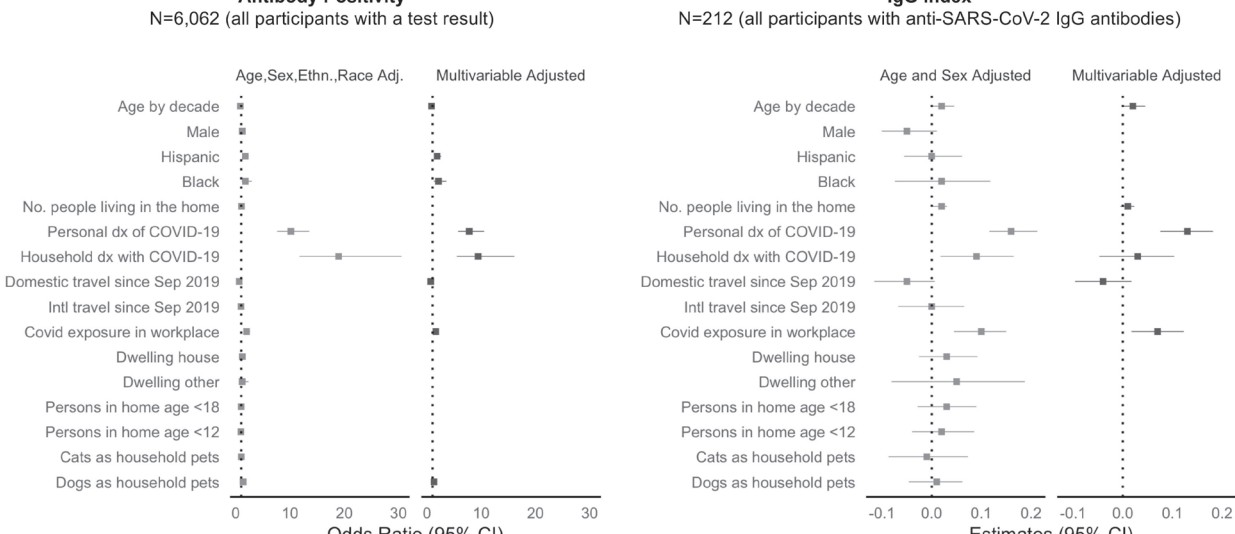

**Figure 3** Potential COVID illness exposure-related factors associated with SARS-CoV-2. dx, diagnosis.

individuals could voluntarily enrol into community screening programmes.[17 18] Given that completely unbiased population scale sampling for seroprevalence studies remains a logistical challenge, we used a sampling approach that involved open enrolment and convenient access to testing facilities made available to all employees working across multiple sites of a large healthcare system; this approach was intended to broadly capture individuals with both patient-related and community-related exposures, while also representative of a relatively wide geographic area in and around Los Angeles County. Although limited to persons who are generally healthy and able to be employed, our study cohort included

individuals representing a diversity of demographic characteristics including ethnicity and race—leading to findings that reflect the disparities that have been persistently observed and reported for COVID-19 infection rates in our local communities. Similar to prior seroprevalence studies conducted across large sample sizes in other regions,[19] results from immunoassays performed at a single time point are likely to underestimate the true prior exposure and infection rate particularly given that SARS-CoV-2 IgG antibody levels are known to wane over a period of weeks to months.[20] Notwithstanding underestimated prior infection rates, related also to variable sensitivity of most IgG immunoassays in relation to timing

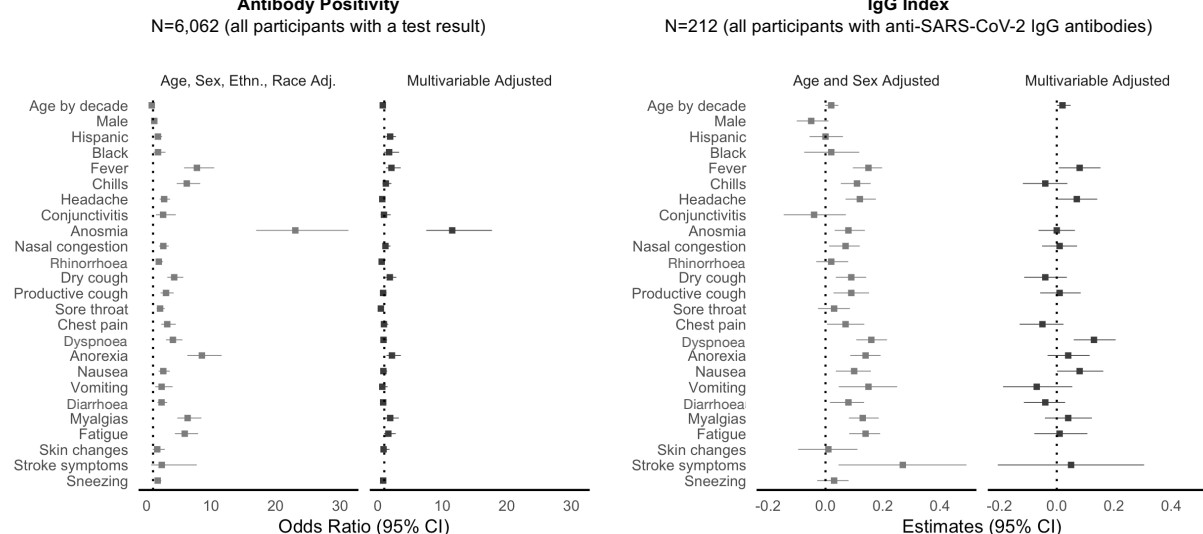

**Figure 4** Potential COVID-19 illness response factors associated with SARS-CoV-2 seroprevalence.

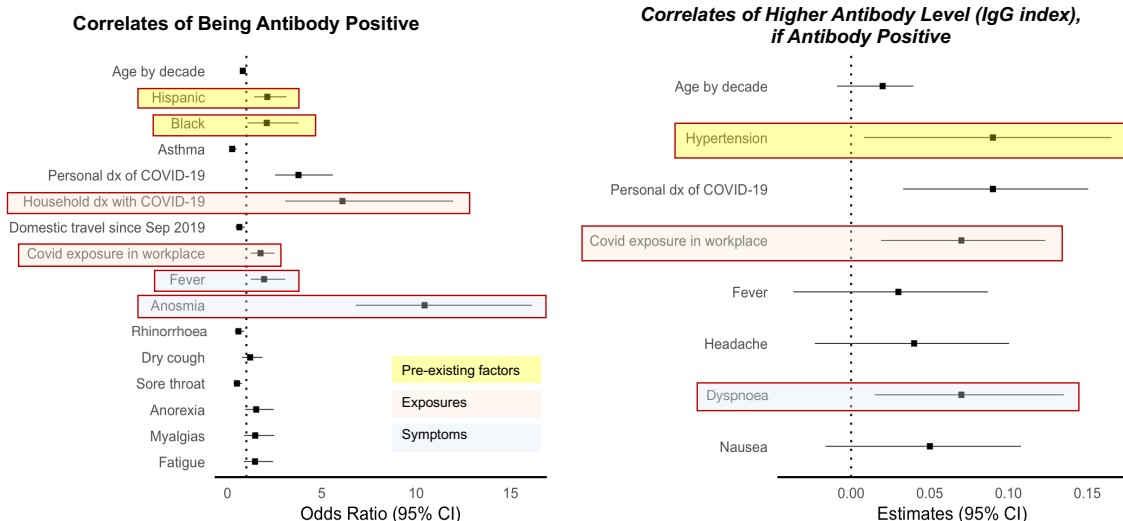

**Figure 5** Factors associated with SARS-CoV-2. dx, diagnosis.

of symptoms (ranging from ≥7 days to 6 months in our study), the overall seroprevalence that we observed is consistent with that reported for regionally proximate populations evaluated during a relatively contemporaneous time period.[21]

Consistent with findings from studies in healthcare workers, seroprevalence patterns in our cohort indicate exposure from the work environment and home environment and likely unmeasured community-based factors.[22] It has been well reported that minority populations, particularly African-Americans and Hispanics, have been disproportionately affected by the COVID-19 pandemic.[23–25] Our study is consistent with these prior findings, but demonstrates that such differences exist even when all participants work not just in the same field, but for the same organisation. Such a finding may indicate that community and non-work-related environmental factors are likely playing a significant role in the spread of COVID-19 among certain minority populations. Even after controlling for a medical diagnosis of COVID-19, African-American race and Hispanic ethnicity remained risk factors for antibody positivity. The persistence of these racial and ethnic disparities may represent structural barriers to care or societally mediated risk. Geographic clustering by race and ethnicity in housing, shopping and social gatherings may be one such factor, while socioeconomic status and ability to self-isolate outside of work likely also contribute.[26–28]

No self-reported pre-existing medical conditions were significantly associated with antibody positivity, indicating that infection itself is agnostic to baseline health. In fact, asthma was negatively associated with the presence of antibodies, or at least antibody levels above the current

threshold we use for positivity. While reactive airway disease is unlikely a protective factor against COVID-19, participants with such conditions may be more likely to diligently follow social distancing guidelines and practise better adherence to hand hygiene and use of personal protective equipment. Hypertension was the only medical condition associated with higher SARS-CoV-2 antibody levels. It remains unclear as to what physiological mechanism may contribute to this finding; however, unmeasured confounding variables, such as medications or renal disease, may function as mediating factors. Further studies will be needed to both verify and elucidate this finding.

Also concordant with prior studies, we found that anosmia was the single strongest symptom associated with SARS-CoV-2 IgG antibody presence.[29–31] Anosmia is recognised as highly specific among the symptoms attributable to COVID-19 and is known to be a particularly frequent finding among younger compared with older infected persons—which likely accounts in part for its especially prominent association with the ability to mount an immune response reflected by degree of detectable seropositivity. Interestingly, neither dyspnoea nor diarrhoea, two commonly cited symptoms, demonstrated a significant association in multivariable analysis.[32 33] This is likely related to the non-specific nature of these symptoms, which are common to multiple viral and non-viral aetiologies. Importantly, dyspnoea was associated with a higher antibody level among those with anti-SARS-CoV-2 antibodies, suggesting that dyspnoea related to COVID-19 may drive a more robust humoral immune response, potentially related to more severe infection. These findings are concordant with the

known phenomenon of proportionate adaptive immune response to higher doses of antigenic stress.[34] The extent to which the generation of measurably higher antibody levels could confer immunity to a larger degree or for a longer duration of time remains unknown. Interestingly, prior studies have demonstrated lower antibody levels among exposed, asymptomatic individuals, a phenomenon which may be attributable to a highly efficient cell-mediated immune response.[35] It has been suggested that higher T-cell levels, whether virus specific or otherwise, may play a role in this finding; however, further research is required.[36 37]

Several limitations of this study merit consideration. Of the employees actively employed at our multisite institution, only a proportion of all eligible participants enrolled; nonetheless, the sample size of the cohort was large, diverse and representative of the source sample.[7] Our seroprevalence estimates were based on using a validated assay of only IgG antibodies; assays of IgM antibodies may offer complementary information in future studies. Data collected on medical history, exposures and symptoms were all self-reported, similar to approaches used in prior studies. We were unable to completely verify prior COVID-19 illness using viral test results in part given lack of universally available testing for all individuals, particularly those with minimal to no symptoms. We observed that history of asthma was associated with lower odds of seropositivity, potentially related to use of corticosteroids or other immunosuppressive therapies; because information on these medications was not available in the current study, they warrant attention in future investigations. Although we collected information on work locations, data regarding specific professions and roles were not consistently captured. Further studies, including potentially training level and seniority of healthcare worker roles, are warranted. Additional details regarding the nature of clinical care provided in certain work areas, particularly those involving nasopharyngeal or respiratory procedures, would also be important for future investigations.

In conclusion, in a highly diverse population of healthcare workers, demographic factors associated with COVID-19 antibody positivity indicate potential factors outside of the workplace are associated with SARS-CoV-2 exposure, although these do not appear related to the number of people or to the presence of children in the home. Further, while dyspnoea may be a marker of more severe disease among those with COVID-19, its presence alone does not indicate infection.

**Author affiliations**
[1]Department of Cardiology, Cedars-Sinai Medical Center, Los Angeles, California, USA
[2]Smidt Heart Institute, Cedars-Sinai Medical Center, Los Angeles, California, USA
[3]F. Widjaja Foundation Inflammatory Bowel and Immunobiology Research Institute, Cedars-Sinai Medical Center, Los Angeles, California, USA
[4]Division of Pulmonary and Critical Care Medicine, University of California, San Diego, La Jolla, California, USA
[5]Departments of Pediatrics, Division of Infectious Diseases and Immunology, and Infectious and Immunologic Diseases Research Center (IIDRC), Department of Biomedical Sciences, Cedars-Sinai Medical Center, Los Angeles, California, USA
[6]Department of Pediatrics, David Geffen School of Medicine at UCLA, Los Angeles, California, USA
[7]Department of Pathology and Laboratory Medicine, Cedars-Sinai Medical Center, Los Angeles, California, USA
[8]Advanced Clinical Biosystems Institute, Department of Biomedical Sciences, Cedars-Sinai Medical Center, Los Angeles, California, USA
[9]Cedars-Sinai Cancer and Department of Medicine, Cedars-Sinai Medical Center, Los Angeles, California, USA
[10]Department of Medicine, Cedars-Sinai Medical Center, Los Angeles, California, USA
[11]Department of Epidemiology, Cedars-Sinai Medical Center, Los Angeles, California, USA
[12]Biobank & Translational Research Core Laboratory, Samuel Oschin Comprehensive Cancer Institute, Cedars-Sinai Medical Center, Los Angeles, California, USA
[13]Department of Public Health Sciences and Comprehensive Cancer Center, University of California, Davis, Davis, California, USA
[14]Department of Medicine and Pharmacology, University of California, San Diego, La Jolla, California, USA
[15]Department of Pharmacology, University of California San Diego School of Medicine, La Jolla, California, USA
[16]Department of Internal Medicine, Division of Hematology, Cedars-Sinai Medical Center, Los Angeles, California, USA
[17]Employee Health Services, Department of Medicine, Cedars-Sinai Medical Center, Los Angeles, California, USA
[18]Brawerman Nursing Institute, Cedars-Sinai Medical Center, Los Angeles, California, USA
[19]Center for Neural Science and Medicine, Department of Biomedical Sciences, Board of Governors Regenerative Medicine Institute, Department of Neurology, Cedars-Sinai Medical Center, Los Angeles, California, USA
[20]David Geffen School of Medicine, University of California, Los Angeles, Los Angeles, California, USA
[21]Chief Medical Officer, Cedars-Sinai Medical Center, Los Angeles, California, USA
[22]La Jolla Institute for Allergy and Immunology, La Jolla, California, USA
[23]Barbra Streisand Women's Heart Center, Cedars-Sinai Medical Center, Los Angeles, California, USA

**Acknowledgements** We are grateful to all the front-line healthcare workers in our healthcare system who continue to be dedicated to delivering the highest quality care for all patients.

**Contributors** JEE, GJB, AHB, PBM, MM, KS, JGB, and SC conceived of the presented idea and contributed to study design. Data collection and analyses were performed by AHB, PB, WH, SJ, EHK, YL, EL, PBM, TTN, KR, MAR, SSt, and NS. JEE, GJB, KS, JGB, and SC led writing of the initial manuscript draft. JEE, GJB, CMA, MAI, MAr, AB, JF-B, JCF, JDG, MH, SKH, MJ, SJ MK, DL, DPBM, AM, NM, PBM, MM, TTN, KR, MAR, CER, RVR, SSh, WGT, JVE, KS, JGB, and SC reviewed, edited and finalised the manuscript, providing critical feedback and changes prior to submission.

**Funding** This work was supported in part by the Cedars-Sinai Medical Center, the Erika J Glazer Family Foundation, the F. Widjaja Family Foundation, the Helmsley Charitable Trust, the NIH/NCI (grant U54-CA260591), and the NIH/ NHLBI (grant K23-HL153888).

**Map disclaimer** The depiction of boundaries on this map does not imply the expression of any opinion whatsoever on the part of BMJ (or any member of its group) concerning the legal status of any country, territory, jurisdiction or area or of its authorities. This map is provided without any warranty of any kind, either express or implied.

**Competing interests** None declared.

**Patient consent for publication** Not required.

**Ethics approval** The study protocol was approved by the Cedars-Sinai Institutional Review Board and all participants provided written informed consent.

**Provenance and peer review** Not commissioned; externally peer reviewed.

**Data availability statement** The data that support the findings of this study are available from Cedars-Sinai Medical Center upon reasonable request. The data

are not publicly available due to the contents including information that could compromise research participant privacy/consent.

**ORCID iD**
Susan Cheng http://orcid.org/0000-0002-4977-036X

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
