## [Reviewer comments · BMJ Open]

ARTICLE DETAILS

TITLE (PROVISIONAL)	Seroprevalence of Antibodies to SARS-CoV-2 in Healthcare Workers: A Cross-Sectional Study
AUTHORS	Ebinger, Joseph; Botwin, Gregory; Albert, Christine; Alotaibi, Mona; Ardit, Moshe; Berg, Anders; Binek, Aleksandra; Botting, Patrick; Fert-Bober, Justyna; Figueiredo, Jane; Grein, Jonathan; Hasan, Wohaib; Henglin, Mir; Hussain, Shehnaz; Jain, Mohit; Joung, Sandy; Karin, Michael; Kim, Elizabeth; Li, Dalin; Liu, Yunxian; Luong, Eric; McGovern, Dermot; Merchant, Akil; Merin, Noah; Miles, Peggy; Minissian, Margo; Nguyen, Trevor Trung; Raedschelders, Koen; Rashid, Mohamad; Riera, Celine; Riggs, Richard; Sharma, Sonia; Sternbach, Sarah; Sun, Nancy; Tourtellotte, Warren; Van Eyk, Jennifer; Sobhani, Kimia; Braun, Jonathan; Cheng, Susan

VERSION 1 – REVIEW

REVIEWER	Lahner Edith Sapienza University of Rome
REVIEW RETURNED	15-Sep-2020

GENERAL COMMENTS	I read with interest the paper on seroprevalence in a large sample of HCWs. The study is well conceived and performed with interesting and detailed statistical analysis. I think that it adds valid and important information on the spread of SARS-CoV2 infection. I would have eventually stressed the predictor anosmia in the conclusions. I have no further comments.
--

REVIEWER	Stefani Thomas University of Minnesota
REVIEW RETURNED	16-Sep-2020

GENERAL COMMENTS	Ebinger and colleagues conducted a study of SARS-CoV-2 antibody seroprevalence among a population of 6,062 adults employed in the multi-site Cedars-Sinai healthcare delivery system in Los Angeles County. Bayesian and multi-variate analyses were used to estimate seroprevalence and the factors associated with seropositivity and antibody titers including pre-existing demographic and clinical characteristics. The authors claim that after adjusting for potential confounding variables, pre-existing medical conditions were not associated with antibody positivity, but seropositivity was associated with younger age, Hispanic ethnicity, and African-American race, as well as the presence of either a personal or household member having a prior diagnosis of COVID-19. Although the main strength of the study is the diverse participant population, the generalizability of the results is likely
---

	narrower than what the authors claim due to the use of various statistical models, the robustness of which is not provided. The following points should be addressed: Major points:  1. pg. 9, Statistical Analyses: Why are the Iterative Proportional Fitting procedure and the Bayesian multilevel hierarchical logistic regression model the most appropriate statistical models for the data presented in this study? 2. pg. 10, Factors Associated with Seroprevalence: “Initial models were deliberately sparse ... and those variables with associations meeting a significance threshold of $P < 0.10$ were advanced for inclusion in a final multivariable model...” 0.10 is a rather low threshold for statistical significance. Were the models tested using a more stringent level of significance, and if so, what was the outcome? 3. pg. 11: It is not clear why there is a lack of consistency with the use of odds ratios or beta when assessing factors that are potentially associate with seropositivity. 4. pg. 11, lines 32-33: History of asthma is listed as one of the main factors associated with lower odds of being seropositive. One could surmise that this is related to the use of corticosteroids or immunosuppressive medication. Did the authors examine the association of prescription corticosteroids and immunosuppressive medication in this population in the context of SARS-CoV-2 seropositivity? 5. pg. 12, line 16: “Dyspnea was significantly associated with titer levels...” The Abbott assay does not report titers. Please correct this statement. 6. pg. 15, last paragraph: The authors should consider eliminating this paragraph. Recent travel, type of home, number of people living in the home, youth or children in the home, and having domestic pets such as cats or dogs are arguably not factors that one would anticipate being associated with an antibody-based measure of SARS-CoV-2 exposure. Alternative variables to consider include recent international travel, or number of people living in the home with a laboratory-confirmed diagnosis of COVID-19. 7. The Supplemental Methods appear to be missing. 8. Supplemental Table 5: Domestic Travel and Dogs in home are shown to be significantly related to SARS-CoV-2 seroprevalence based on Model 1; however, this is contrary to what is shown in Figure 3 and Figure 5. Minor comments  1. Change all mentions of “Covid-19” to “COVID-19” as COVID is an acronym. 2. pg. 8, Serologic Assays: Indicate the specimen type(s) used for the SARS-CoV-2 immunoassay. Relatedly, what were the transport conditions for the specimens after collection and prior to their arrival in the laboratory for testing? 3. pg. 11, line 50: Clarify what is meant by “working in a location where COVID-19 patients are treated”. Does this mean employees who are directly involved in patient care, or does it also include support staff in non-clinical roles? 4. Fig. 2, 3, and 4: Correct the titles of these plots to Antibody Index instead of IgG Titer Index. 5. Supplemental Table 1 and 2: Add a column with the % Sensitivity (Table 1) and % Specificity (Table 2).
--	---

REVIEWER	Kohei Fujita National Hospital Organization Kyoto Medical Center, Kyoto, Japan
REVIEW RETURNED	03-Oct-2020

GENERAL COMMENTS	This study provides the excellent information of the SARS-CoV-2 seroprevalence in Los Angeles. Global pandemic situation suggests the existence of racial differences of morbidity and mortality caused by COVID-19. This study revealed the racial differences of SARS-CoV-2 seroprevalence. Furthermore, The authors showed that specific ethnicity were associated with antibody positivity. It is also interesting results that dwelling type, number of people and having children were not associated with either seroprevalence or antibody titer. I have some minor comments to the authors.  1. Because this study targets healthcare workers, the readers will interest the seroprevalence with respect to each profession (doctor, nurse or medical clerk, etc.). Will you have any data? 2. In addition to Q1, I want to know the seroprevalence with respect to each medical section (for example, it is important factor whether working at emergency medicine district, respiratory medicine district or otolaryngology medicine district, where, I think, having high risk of SARS-CoV-2 exposure) 3. In the Table 1, the term of "Immune" in the characteristics is unclear. Immunosuppressive status? Having autoimmune disease? Please clarify. 4. Obesity is important risk factor for COVID-19. How about adding to analysis?
--

REVIEWER	Adrian Shields University of Birmingham
REVIEW RETURNED	09-Oct-2020

GENERAL COMMENTS	Thank you for the opportunity to review this manuscript. In this study, Ebinger and colleagues have tested 6,000 health care workers at Cedar-Sanai Medical Center, San Diego for antibodies against the nucleocapsid of SARS-CoV-2 to generate an estimate of seroprevalence across their employee base. I have the following comments:  1. Verification and performance characteristics of the Abbott assay As far as I can tell, the local verification of the performance characteristics of this assay are based on 60 cases of COVID-19 and 178 control samples. Based on that sample, the assay has 88.3% sensitivity and 100% specificity when used by the Cedar-Sanai laboratory. I am unable to tell what population was used to source these positive samples. The following information is needed to understand the performance of the test used:
---

a) What population were the positive control samples used to determine sensitivity derived from; were they cases that were severe enough to be hospitalised or non-hospitalised cases?
b) What criteria were used to determine a positive "case". Was PCR positivity used as the reference for determining the positive cases (if so, what PCR assay was used for purpose and its own performance characteristics)
c) What internal QC has been used for this assay? What is the uncertainty of measurement for the assay? Has internal quality control material, has been designed to run close to assay's cut-off for positivity. What is %CV for the iQC material this and is the %CV linear across the assay's measurement range? Was there an equivocal range?

2. Validation of the performance characteristics of the Abbott assay for the target population

It is not clear from the methods, whether the laboratory has independently undertaken a verification of the assay with respect to its performance characteristics for the target population.

Looking at the data presented, 634 individuals appeared to have received a personal diagnosis of COVID-19. Were that diagnosis reached by PCR positivity? Of those 634 individuals, only 104 individuals had a positive antibody test (16%). This is considerably different to the 88.3% sensitivity suggested by the verification process.

One explanation might be that the verification of assay performance was conducted on samples derived from hospitalised patients, who have severe disease and a more easily detectable serological response. However, the assay has been used to try and identify individuals with a modest antibody response because they have had mild disease.

Concerns have been raised regarding the performance of the Abbott assay by other groups; [https://doi.org/10.1016/S0140-6736\(20\)31830-4](https://doi.org/10.1016/S0140-6736(20)31830-4)

Concerns regarding the design of immunoassays leading to underestimation of seroprevalence have also been raised: <https://www.bmj.com/content/370/bmj.m3364>

3. Time from symptom onset.

Analysis is restricted to those who gave samples at least 7 days from symptom onset, which is entirely reasonable considering the study only looks at the IgG response. However, the authors reference Bryan et al - they state the sensitivity of the Abbott assay is only 53.1% (95% confidence interval [CI], 39.4% to 66.3%) at 7 days.

What was the distribution of time from symptom onset. Furthermore, do IgG levels directed against the NP reduce with time; the Bryan paper looks at 17 days post symptom onset, but have the authors extended this further? IgG levels do reduced with time from symptom onset, so how long are they detectable with using the Abbott platform?

	4. General comments The regression modelling seems secure, however would benefit from review by an expert statistician. The conclusions drawn regarding increased risk of seropositivity in individuals of African American and Hispanic ethnicity are relevant and broadly concordant with other studies. However, taking the above into consideration and the other work that has raised concerns regarding the performance of IgG based NP assays, clarity is needed with respect to how well the assay performs in order to interpret the downstream analysis. Minor points 1. It's worth stating that the laboratory has (I think) used plasma rather than serum to determine seropositivity in this work.
--	---

VERSION 1 – AUTHOR RESPONSE

Comments from Reviewer 1

1. I read with interest the paper on seroprevalence in a large sample of HCWs. The study is well conceived and performed with interesting and detailed statistical analysis. I think that it adds valid and important information on the spread of SARS-CoV2 infection. I would have eventually stressed the predictor anosmia in the conclusions. I have no further comments.

Reply: We appreciate the Reviewer's valuable comments and suggestion. As advised, we have added additional emphasis regarding the association of Anosmia in the conclusion:

Page 15: "Also concordant with prior studies, we found that anosmia was the single strongest symptom associated with SARS-CoV-2 IgG antibody presence.²⁹⁻³¹ Anosmia is recognized as not only highly specific among the symptoms attributable to COVID-19 but is also known to be a particularly frequent finding among younger compared to older infected persons – which likely accounts in part for its especially prominent association with the ability to mount an immune response reflected by degree of detectable seropositivity."

1

bmjopen-20200-43584

Comments from Reviewer 2

1. pg. 9, Statistical Analyses: Why are the Iterative Proportional Fitting procedure and the Bayesian multilevel hierarchical logistic regression model the most appropriate statistical models for the data presented in this study?

Reply: We thank the Reviewer for the helpful comments and suggestions provided. We especially appreciate the queries regarding the statistical approaches employed. Following a careful review of analytical methods used for study designs similar to ours, we selected the iterative proportional fitting procedure based on its ability to accommodate variation in results that could be introduced by potential sampling bias. Recognizing that our study cohort included a large number, yet not the total number, of all eligible healthcare workers employed in our health system, we sought a method that could accommodate for possible sampling bias – and consistent with methods that have been applied in prior as well as contemporary studies related to SARS-CoV-2 exposure (e.g. MMWR 2020;69:960-964 PubMed). In addition to accounting for potential bias from sampling, we also recognized the need to account for variation in results that could be introduced by potential bias related to the sensitivity and specificity of the antibody test itself. To this end, we applied the Bayesian multilevel hierarchical logistic regression model given its ability to accommodate detailed information that we collated from all available sources that had previously reported sensitivity and specificity of the Abbott Architect assay for SARS-CoV-2 IgG when used in different types of populations. We applied this Bayesian modeling according to similar approaches that have been reported in prior as well as contemporary studies that have evaluated methods for estimating seroprevalence of SARS-CoV-2 antibodies in different settings (e.g. JAMIA Open 2020; doi 10.1093 and J Adolesc Health 2020;67:763-768 PubMed). As helpfully suggested by the Reviewer, we have now added clarification of these important details to the revised manuscript:

Page 9: “Given that our study cohort included a large number, yet not the total number, of all eligible healthcare workers employed in our health system, we used the iterative proportional fitting (IPF) procedure to account for any possible sampling bias; notably, the IPF has been applied effectively in prior as well as contemporary studies related to SARS-CoV-2 exposure.¹¹ Accordingly, we integrated source population-level demographic data, representative of the entire Cedars-Sinai employee base, with data from our enrolled study sample and then used IPF to estimate the number of eligible employees within each demographic category (with provided population totals considered the target, using constraints derived from our sample).¹² In addition to accounting for potential bias from sampling, we also recognized the need to account for potential bias related to the previously reported sensitivity and specificity of the antibody assay (). Thus, in accordance with methods applied in similar seroprevalence studies,^{13,14} we fit a Bayesian multilevel hierarchical logistic regression model using RStan,^{15,16} including reported age, gender, race/ethnicity and site as coefficients, to model exposure probability. We then estimated the seroprevalence within each post-stratified demographic category based on the averaged and weighted value of the expected number of employees within that category.”

2. pg. 10, Factors Associated with Seroprevalence: “Initial models were deliberately sparse ... and those variables with associations meeting a significance threshold of $P < 0.10$ were advanced for inclusion in a final multivariable model....” 0.10 is a rather low threshold for statistical significance. Were the models tested using a more stringent level of significance, and if so, what was the outcome?

Reply: We agree with the Reviewer that we had originally selected a more liberal threshold of statistical significance ($P < 0.10$) for considering variables that could be advanced from the sparse models to the full multivariable adjusted models, in order to increase the chance of detecting potentially important associations. As advised, we have now repeated all of our analyses using the more stringent threshold of $P < 0.05$ and the results are overall similar to those of the original analyses. Given the similarity of results, we have now updated all reported values to reflect results from these more stringent analyses:

Page 10: “Initial models were deliberately sparse, adjusting for a limited number of key covariates (e.g. age, gender) and those variables with associations meeting a significance threshold of

P<0.05 were advanced for inclusion in a final multivariable model along with only other variables identified as significant from the sparse regressions.”

3. pg. 11: It is not clear why there is a lack of consistency with the use of odds ratios or beta when assessing factors that are potentially associated with seropositivity.

Reply: We appreciate this comment and, for all updated analyses, we have now carefully re-checked all displayed values for odds ratios and beta estimates to confirm concordance and consistency for all reported results. We have also added clarification of which values are the results of logistic models for the binary outcome of seropositivity and which values are the results of linear models for the continuous outcome of IgG index level:

Page 10: “A final separate logistic or linear multivariable model was constructed for each of the 3 categories of variables in relation to the binary outcome of seropositivity or the continuous outcome of IgG antibody level, respectively.”

4. pg. 11, lines 32-33: History of asthma is listed as one of the main factors associated with lower odds of being seropositive. One could surmise that this is related to the use of corticosteroids or immunosuppressive medication. Did the authors examine the association of prescription corticosteroids and immunosuppressive medication in this population in the context of SARS-CoV-2 seropositivity?

Reply: We appreciate this query and were unable to include prescription corticosteroids or immunosuppressive medication use given that these data were not collected via our survey report and thus unavailable for analyses. We have now added report of this limitation to the revised manuscript:

Page 16: “We observed that history of asthma was associated with lower odds of seropositivity, potentially related to use of corticosteroids or other immunosuppressive therapies; because information on these medications was not available in the current study, they warrant attention in future investigations.”

5. pg. 12, line 16: “Dyspnea was significantly associated with titer levels....” The Abbott assay does not report titers. Please correct this statement.

Reply: We thank the Reviewer for this comment. We have now corrected the statement in the revised manuscript, as helpfully advised:

Page 12: “Dyspnea was significantly associated with higher IgG index levels....”

6. pg. 15, last paragraph: The authors should consider eliminating this paragraph. Recent travel, type of home, number of people living in the home, youth or children in the home, and having domestic pets such as cats or dogs are arguably not factors that one would anticipate being associated with an antibody-based measure of SARS-CoV-2 exposure. Alternative variables to consider include recent international travel, or number of people living in the home with a laboratory-confirmed diagnosis of COVID-19.

Reply: We appreciate this suggestion and have removed this paragraph from the revised manuscript, as advised.

7. The Supplemental Methods appear to be missing.

Reply: We apologize for the typographical error that suggested presence of a Supplemental Methods. Given that all methodological details are included in the main manuscript, we have removed any reference to a Supplemental Methods section from the revised manuscript.

8. Supplemental Table 5: Domestic Travel and Dogs in home are shown to be significantly related to SARS-CoV-2 seroprevalence based on Model 1; however, this is contrary to what is shown in Figure 3 and Figure 5.

Reply: We agree that the certain factors, such as domestic travel and dog ownership, appears significantly associated with seropositivity in the sparse models and then these associations became attenuated in the more fully adjusted multivariable models with results depicted in the figures. We have now added clarification to provide context for these results that indicate likely presence of unmeasured confounders (e.g. higher socioeconomic status):

Pages 11-12: “Notably, domestic travel, dwelling type, number of people in the home, and having children or common domestic pets were not associated with either seroprevalence or antibody level in the more completely adjusted multivariable models, which can account at least partially for the effects unmeasured confounders that are not captured in the sparser models.”

Minor comments:

1. Change all mentions of “Covid-19” to “COVID-19” as COVID is an acronym.

Reply: We appreciate this suggestion and have made this edit throughout the revised manuscript.

2. pg. 8, Serologic Assays: Indicate the specimen type(s) used for the SARS-CoV-2 immunoassay. Relatedly, what were the transport conditions for the specimens after collection and prior to their arrival in the laboratory for testing?

Reply: We thank the Reviewer for these important suggestions. We have now included details regarding the use of plasma biospecimens that were transported within 1 hour of collection to our clinical chemistry laboratory for immediate testing:

Page 7: “For all participants, EDTA plasma specimens were transported within 1 hour of phlebotomy to the Cedars-Sinai Department of Pathology and Laboratory Medicine and underwent serology testing using the Abbott Diagnostics SARS-CoV-2 IgG chemiluminescent microparticle immunoassay...”

3. pg. 11, line 50: Clarify what is meant by “working in a location where COVID-19 patients are treated”. Does this mean employees who are directly involved in patient care, or does it also include support staff in non-clinical roles?

Reply: We appreciate this important query. We have now added details regarding the reported work locations, particularly those related to higher levels of COVID-19 exposure:

Page 7: “Work location was specified as spending most working hours in an ICU (COVID-19 or non-COVID-19 designated), non-ICU ward (COVID-19 or non-COVID-19 designated), outpatient clinic, office, work-from-home, or other location.”

4. Fig. 2, 3, and 4: Correct the titles of these plots to Antibody Index instead of IgG Titer Index.

Reply: We thank the Reviewer for this important comment and have changed all occurrences of the term 'titer' to the more correct term 'index'.

5. Supplemental Table 1 and 2: Add a column with the % Sensitivity (Table 1) and % Specificity (Table 2).

Reply: We appreciate this suggestion and have now added columns for sensitivity and specificity to these respective supplemental tables, as helpfully advised.

1

bmjopen-20200-43584

Comments from Reviewer 3

This study provides the excellent information of the SARS-CoV-2 seroprevalence in Los Angeles. Global pandemic situation suggests the existence of racial differences of morbidity and mortality caused by COVID-19. This study revealed the racial differences of SARS-CoV-2 seroprevalence. Furthermore, The authors showed that specific ethnicity were associated with antibody positivity. It is also interesting results that dwelling type, number of people and having children were not associated with either seroprevalence or antibody titer. I have some minor comments to the authors.

Reply: We thank the Reviewer for the valuable comments and suggestions provided.

1. Because this study targets healthcare workers, the readers will interest the seroprevalence with respect to each profession (doctor, nurse or medical clerk, etc.). Will you have any data?

Reply: We appreciate this important comment and were unable to systematically collect data on the specific role or profession of each healthcare worker. We have now added mention of this important detail to the limitations section of the revised manuscript:

Page 16: "Although we collected information on work locations, data regarding specific professions and roles were not consistently captured. Further studies, including potentially training level and seniority of healthcare worker roles, are warranted."

2. In addition to Q1, I want to know the seroprevalence with respect to each medical section (for example, it is important factor whether working at emergency medicine district, respiratory medicine district or otolaryngology medicine district, where, I think, having high risk of SARS-CoV-2 exposure)

Reply: We completely agree with the Reviewer that specific areas of primary work function are likely to confer higher or lower risk for SARS-CoV-2 exposure. Although we were able to capture data on whether the primary work area involved patient care, management of COVID-19 patients, and level of care, we were unable to systematically collect more specific data on location and area of primary work function. We have now added mention of this important detail to the limitations section of the revised manuscript:

Page 16: "Additional details regarding the nature of clinical care provided in certain work areas, particularly those involving nasopharyngeal or respiratory procedures, would also be important for future investigations."

3. In the Table 1, the term of "Immune" in the characteristics is unclear. Immunosuppressive status? Having autoimmune disease? Please clarify.

Reply: We thank the Reviewer for identifying this typographical error. We apologize for the lack of clarity regarding the "immune" variable term, which was intended to denote autoimmune disease. We have now revised the manuscript to clarify that the data presented are for participants who reported a medical history of autoimmune disease.

4. Obesity is important risk factor for COVID-19. How about adding to analysis?

Reply: We thank the Reviewer for this important suggestion. We have now integrated data on height and weight so that we can calculate body mass index and generate the obesity variable for inclusion in all analyses. As now reported in the new Supplemental Table 4, the association of obesity with IgG seropositivity was OR 0.82 (95% CI 0.55, 1.24; P=0.35) and with IgG index was beta=0.01 per 10 units (SE 0.04; P=0.71). As helpfully suggested by the Reviewer, we have now updated all of our analyses in the revised manuscript to include accounting for obesity.

1

bmjopen-20200-43584

Comments from Reviewer 4

Thank you for the opportunity to review this manuscript. In this study, Ebinger and colleagues have tested 6,000 health care workers at Cedar-Sinai Medical Center, San Diego for antibodies against the nucleocapsid of SARS-CoV-2 to generate an estimate of seroprevalence across their employee base. I have the following comments:

Reply: We thank the Reviewer for the valuable comments and suggestions provided.

1. Verification and performance characteristics of the Abbott assay

As far as I can tell, the local verification of the performance characteristics of this assay are based on 60 cases of COVID-19 and 178 control samples. Based on that sample, the assay has 88.3% sensitivity and 100% specificity when used by the Cedar-Sinai laboratory. I am unable to tell what population was used to source these positive samples.

The following information is needed to understand the performance of the test used:

- a) What population were the positive control samples used to determine sensitivity derived from; were they cases that were severe enough to be hospitalised or non-hospitalised cases?
- b) What criteria were used to determine a positive "case". Was PCR positivity used as the reference for determining the positive cases (if so, what PCR assay was used for purpose and its own performance characteristics)

c) What internal QC has been used for this assay? What is the uncertainty of measurement for the assay? Has internal quality control material, has been designed to run close to assay's cut-off for positivity. What is %CV for the iQC material this and is the %CV linear across the assay's measurement range? Was there an equivocal range?

Reply: We appreciate these very thoughtful and important queries from the Reviewer. We have now added details regarding the procedures used to verify local performance characteristics of the assay. Specifically, we used samples obtained from 60 hospitalized cases of clinical COVID-19 illness that were identified based on presence of a positive PCR assay (RT-qPCR assay based on A*STAR Fortitude Kit 2.0). Similarly, the 178 hospitalized controls were identified based on non-COVID-19 illness and a negative PCR assay. All cases and controls were from patients hospitalized between March 2020 and May 2020. The time lapsed between symptom onset and antibody assay was ~7 to 14 days overall for our cases, and this timing is similar to the 8 to 13 day time lapse reported by Abbott from its own performance testing data. Accordingly, our local performance data analysis revealed a sensitivity or positive percent agreement (PPA) of 88.3%, which is similar to the Abbott result of 86.4% within their 8 to 13 day time period (along with recognition that the PPA approaches 100% at ≥ 14 days following symptom onset). The QC procedures that we used for the Abbott Architect assay were based on the QC material provided by Abbott. The CVs are $\leq 1.4\%$ for positive and negative controls. Whereas internal QC is considered relevant to cartridge based lateral flow assays and mass spectrometry assays, the recommended QC procedures for the Abbott Architect assay involve incorporating external controls or assayed QC material that is treated like patient samples (i.e. not added to patients samples or built in as part of the assay). The positive QC runs jst about 3.00 index and the negative is essentially a blank plasma based matrix and runs at 0.00 index. As thoughtfully suggested by the Reviewer, we have now added these important details to the revised manuscript:

Page 8: "To verify local performance of the assay, we used samples obtained at our institution from 60 cases of COVID-19 (hospitalized between March and May 2020) and 178 controls that were identified based on positive or negative PCR assay (RT-qPCR assay based on A*STAR Fortitude Kit 2.0) with a time lapse between symptom onset and antibody assay of ~7 to 14 days. We found a sensitivity or positive percent agreement (PPA) of 88.3%, with CVs of $\leq 1.4\%$ for positive and negative controls."

2. Validation of the performance characteristics of the Abbott assay for the target population

It is not clear from the methods, whether the laboratory has independently undertaken a verification of the assay with respect to its performance characteristics for the target population.

Looking at the data presented, 634 individuals appeared to have received a personal diagnosis of COVID-19. Were that diagnosis reached by PCR positivity? Of those 634 individuals, only 104 individuals had a positive antibody test (16%). This is considerably different to the 88.3% sensitivity suggested by the verification process.

One explanation might be that the verification of assay performance was conducted on samples derived from hospitalised patients, who have severe disease and a more easily detectable serological response. However, the assay has been used to try and identify individuals with a modest antibody response because they have had mild disease.

Concerns have been raised regarding the performance of the Abbott assay by other groups; [https://urldefense.com/v3/__https://doi.org/10.1016/S0140-6736\(20\)31830-4__;!!K0mNBZxC8_2BBQ!gkvQ0YGE4LqYyLHyTn_-lxOYCp17ITWcbK-4IzguG4aNUBIQ6gC7qvY1qJjrKUtHiQ\\$](https://urldefense.com/v3/__https://doi.org/10.1016/S0140-6736(20)31830-4__;!!K0mNBZxC8_2BBQ!gkvQ0YGE4LqYyLHyTn_-lxOYCp17ITWcbK-4IzguG4aNUBIQ6gC7qvY1qJjrKUtHiQ$)

Concerns regarding the design of immunoassays leading to underestimation of seroprevalence have also been raised:

[https://urldefense.com/v3/___https://www.bmj.com/content/370/bmj.m3364___!!KOmnBZxC8_2BBQ!gk vQ0YGE4LqYyLHyTn_-lxOYCp17ITWcbK-4IzguG4aNUBIQ6gC7qvY1qJjbqC5aTA\\$](https://urldefense.com/v3/___https://www.bmj.com/content/370/bmj.m3364___!!KOmnBZxC8_2BBQ!gk vQ0YGE4LqYyLHyTn_-lxOYCp17ITWcbK-4IzguG4aNUBIQ6gC7qvY1qJjbqC5aTA$)

Reply: We completely agree with the Reviewer on this very important point. As has been raised by prior reports, including those identified above, there are multiple possible sources of discordance between the reported COVID-19 diagnosis rate and the detected IgG seroprevalence rate. While recognizing that limits in assay performance are one potential source, another major source is the time lapsed between prior COVID-19 diagnosis and the date of the blood draw for the assay. In fact, we observed that of all persons who reported a prior COVID-19 diagnosis, a majority (62%) also reported the onset of their symptoms attributable to COVID-19 as having occurred >1-3 months prior to the date of the assay blood draw. Thus, the lower than expected seroprevalence rate that we observed is particularly consistent with prior reports of waning prevalence of peripherally circulating SARS-CoV-2 antibodies detectable in previously infected individuals who are serially assessed over time (JAMA. 2020;324:1279-1281). Given the importance of the Reviewer's astute observation, we have added clarification of these details to the revised manuscript:

Page 14: "Similar to prior seroprevalence studies conducted across large samples sizes in other regions,¹⁹ results from immunoassays performed at a single timepoint are likely to underestimate the true prior exposure and infection rate particularly given that SARS-CoV-2 IgG antibody levels are known to wane over a period of weeks to months.²⁰"

3. Time from symptom onset.

Analysis is restricted to those who gave samples at least 7 days from symptom onset, which is entirely reasonable considering the study only looks at the IgG response. However, the authors reference Bryan et al - they state the sensitivity of the Abbott assay is only 53.1% (95% confidence interval [CI], 39.4% to 66.3%) at 7 days.

What was the distribution of time from symptom onset. Furthermore, do IgG levels directed against the NP reduce with time; the Bryan paper looks at 17 days post symptom onset, but have the authors extended this further? IgG levels do reduced with time from symptom onset, so how long are they detectable with using the Abbott platform?

Reply: We completely agree with the Reviewer on the extremely important point of needing to clarify the timing with respect to the two procedures described in this study: (i) internal testing in the QC study, and (ii) implementation in seroprevalence study. For internal testing performed as part of our QC procedures, the time lapsed between symptom onset and blood draw for the assay was ~7 to 14 days, as described above. For the seroprevalence study, the time lapse between symptom onset and blood draw for the assay was much more variable and ranged from 7 days to 6 months; this variability in timing is similar to that described in prior reports of seroprevalence studies performed across a range of community-based as well as healthcare worker cohorts (e.g. JAMA 2020;324:893-895 PubMed). Given the importance of the Reviewer's very astute observation, we have added clarification of these details to the revised manuscript:

Page 14: "Notwithstanding underestimated prior infection rates, related also to variable sensitivity of most IgG immunoassays in relation to timing of symptoms (ranging from ≥ 7 days to 6 months in our study), the overall seroprevalence that we observed is consistent with that reported for regionally proximate populations evaluated during a relatively contemporaneous time period.²¹"

4. General comments

The regression modelling seems secure, however would benefit from review by an expert statistician. The conclusions drawn regarding increased risk of seropositivity in individuals of African American and Hispanic ethnicity are relevant and broadly concordant with other studies. However, taking the above into consideration and the other work that has raised concerns regarding the performance of IgG based NP assays, clarity is needed with respect to how well the assay performs in order interpret the downstream analysis.

Reply: We completely agree with the Reviewer on the important point of ensuring rigorous statistical methods to account for variability in assay performance due to technical issues as well as due to factors intrinsic to study design as well as characteristics of the study population. For this reason, we carefully reviewed previously published reports of both methods and results of comparable seroprevalence studies and applied an approach that incorporates statistical accommodation for both variation potentially derived by the assay as well as variation potentially derived from population sampling. In particular, that our study cohort included a large number, yet not the total number, of all eligible healthcare workers employed in our health system, we sought a method that could accommodate for possible sampling bias – and consistent with methods that have been applied in prior as well as contemporary studies related to SARS-CoV-2 exposure (e.g. MMWR 2020;69:960-964 PubMed). In addition to accounting for potential bias from sampling, we also recognized the need to account for variation in results that could be introduced by potential bias related to the sensitivity and specificity of the antibody test itself. Thus, we applied the Bayesian multilevel hierarchical logistic regression model given its ability to accommodate detailed information that we collated from all available sources that had previously reported sensitivity and specificity of the Abbott Architect assay for SARS-CoV-2 IgG when used in different types of populations. We applied this Bayesian modeling according to similar approaches that have been reported in prior as well as contemporary studies that have evaluated methods for estimating seroprevalence of SARS-CoV-2 antibodies in different settings (e.g. JAMIA Open 2020; doi 10.1093 and J Adolesc Health 2020;67:763-768 PubMed). We have now included clarifying details regarding these statistical methods in the revised manuscript:

Page 9: “Given that our study cohort included a large number, yet not the total number, of all eligible healthcare workers employed in our health system, we used the iterative proportional fitting (IPF) procedure to account for any possible sampling bias; notably, the IPF has been applied effectively in prior as well as contemporary studies related to SARS-CoV-2 exposure.¹¹ Accordingly, we integrated source population-level demographic data, representative of the entire Cedars-Sinai employee base, with data from our enrolled study sample and then used IPF to estimate the number of eligible employees within each demographic category (with provided population totals considered the target, using constraints derived from our sample).¹² In addition to accounting for potential bias from sampling, we also recognized the need to account for potential bias related to the previously reported sensitivity and specificity of the antibody assay (Supplemental Tables 1-2). Thus, in accordance with methods applied in similar seroprevalence studies,^{13,14} we fit a Bayesian multilevel hierarchical logistic regression model using RStan,^{15,16} including reported age, gender, race/ethnicity and site as coefficients, to model exposure probability. We then estimated the seroprevalence within each post-stratified demographic category based on the averaged and weighted value of the expected number of employees within that category.”

Minor points:

1. It's worth stating that the laboratory has (I think) used plasma rather than serum to determine seropositivity in this work.

Reply: The Reviewer is absolutely correct and we agree with this important point. We have now added clarification that EDTA plasma was use for all assays in the study.

Page 7: “For all participants, EDTA plasma specimens were transported within 1 hour of phlebotomy to the Cedars-Sinai Department of Pathology and Laboratory Medicine and underwent serology testing using the Abbott Diagnostics SARS-CoV-2 IgG chemiluminescent microparticle immunoassay (Abbott Diagnostics, Abbott Park, IL) performed on an Abbott Diagnostics Architect ci16200 analyzer.”

VERSION 2 – REVIEW

REVIEWER	Stefani Thomas University of Minnesota
REVIEW RETURNED	01-Dec-2020

GENERAL COMMENTS	The authors have provided thoughtful and comprehensive responses to all of the comments raised by the Editor and the four reviewers, and the manuscript has been revised accordingly. The quality of the manuscript has been improved. I do not have any additional suggestions for revision.
---

REVIEWER	Kohei Fujita Division of Respiratory Medicine, Center for Respiratory Diseases, National Hospital Organization Kyoto Medical Center, Kyoto, Japan
REVIEW RETURNED	01-Dec-2020

GENERAL COMMENTS	Good response.
----------------

REVIEWER	Adrian Shields University of Birmingham
REVIEW RETURNED	01-Dec-2020

GENERAL COMMENTS	Many thanks for the opportunity to review this revised manuscript and to the authors for their consideration of the points raised in my original review; these have been addressed. I believe this manuscript makes an important contribution to our understanding of SARS-CoV-2 risk in health care workers and have recommended this manuscript for publication.
--